# Deciphering neuronal deficit and protein profile changes in human brain organoids from patients with creatine transporter deficiency

Léa Broca-Brisson[1], Rania Harati[2,3], Clémence Disdier[4], Orsolya Mozner[5], Romane Gaston-Breton[1], Auriane Maïza[1], Narciso Costa[1], Anne-Cécile Guyot[1], Balazs Sarkadi[5], Agota Apati[5], Matthew R Skelton[6], Lucie Madrange[7], Frank Yates[7], Jean Armengaud[8], Rifat Hamoudi[9,10,11], Aloïse Mabondzo[1]*

[1]Université Paris-Saclay, CEA, INRAE, Département Médicaments et Technologies pour la Santé, Gif sur Yvette, France; [2]Department of Pharmacy Practice and Pharmacotherapeutics, College of Pharmacy, University of Sharjah, Sharjah, United Arab Emirates; [3]Sharjah Institute for Medical Research, University of Sharjah, Sharjah, United Arab Emirates; [4]CERES BRAIN Therapeutics, Paris, France; [5]Institute of Enzymology, Research Centre for Natural Sciences, ELKH, and Doctoral School of Molecular Medicine, Semmelweis University, Budapest, Hungary; [6]Department of Pediatrics, University of Cincinnati College of Medicine and Division of Neurology, Cincinnati Children's Research Foundation, Cincinnati, United States; [7]SupBiotech/Service d'Etude des Prions et des Infections Atypiques (SEPIA), Institut François Jacob, CEA, Université Paris Saclay, Paris, France; [8]Université Paris-Saclay, CEA, INRAE, Département Médicaments et Technologies pour la Santé (DMTS), SPI, Bagnols-sur-Cèze, France; [9]Clinical Sciences Department, College of Medicine, University of Sharjah, Sharjah, United Arab Emirates; [10]Division of Surgery and Interventional Science, University College London, London, United Kingdom; [11]ASPIRE Precision Medicine Research Institute Abu Dhabi, University of Sharjah, Sharjah, United Arab Emirates

*For correspondence:
aloise.mabondzo@cea.fr

**Abstract** Creatine transporter deficiency (CTD) is an X-linked disease caused by mutations in the SLC6A8 gene. The impaired creatine uptake in the brain results in intellectual disability, behavioral disorders, language delay, and seizures. In this work, we generated human brain organoids from induced pluripotent stem cells of healthy subjects and CTD patients. Brain organoids from CTD donors had reduced creatine uptake compared with those from healthy donors. The expression of neural progenitor cell markers SOX2 and PAX6 was reduced in CTD-derived organoids, while GSK3β, a key regulator of neurogenesis, was up-regulated. Shotgun proteomics combined with integrative bioinformatic and statistical analysis identified changes in the abundance of proteins associated with intellectual disability, epilepsy, and autism. Re-establishment of the expression of a functional SLC6A8 in CTD-derived organoids restored creatine uptake and normalized the expression of SOX2, GSK3β, and other key proteins associated with clinical features of CTD patients. Our brain organoid model opens new avenues for further characterizing the CTD pathophysiology and supports the concept that reinstating creatine levels in patients with CTD could result in therapeutic efficacy.

## eLife assessment

This is an **important** study highlighting how a single protein transporter dysfunction can significantly alter brain biochemistry, potentially playing a crucial role in the intellectual disability in creatine transporter deficiency (CTD) patients. The evidence is **compelling** that the new in vitro CTD model using CTD patient's brain organoid cultures will be widely applicable. Despite minor areas for further exploration, the study significantly enhances our understanding of CTD, offering potential therapeutic targets and a robust foundation for continued research in the field.

## Introduction

CTD is a devastating neurological disorder that results in moderate to severe intellectual disability (ID), epilepsy, and a lack of language development (*Cecil et al., 2001*; *deGrauw et al., 2003*; *Farr et al., 2020*; *Salomons et al., 2003*). Mutations in the *SLC6A8* gene reduce or eliminate creatine transporter (CRT, SLC6A8) production and impair creatine (Cr) uptake, preventing brain Cr accumulation. Individuals with CTD may have mild generalized muscular atrophy, dysmorphic facial features, microcephaly, and gastrointestinal disturbances (*Stockler et al., 2007*; *van de Kamp et al., 2014*). This neglected disorder has been estimated to cause between 1–3% of all X-linked ID (*van de Kamp et al., 2014*) and about 1% of cases with ID of unknown etiology (*Clark et al., 2006*). Currently, there is no efficient treatment for CTD, and the affected males require lifelong familial or institutional care, representing a significant economic burden for both the individual and the society (*Fernandes-Pires and Braissant, 2022*).

The lack of functional CRT in the brain represents a significant roadblock in improving the clinical outcome of CTD patients (*Ohtsuki et al., 2002*; *van de Kamp et al., 2014*). Several combinations of nutritional supplements or Cr precursors l-arginine and l-glycine, have been studied as therapeutic approaches for CTD, but they have shown limited success (*Bruun et al., 2018*; *Valayannopoulos et al., 2013*), compelling development of alternative strategies for the treatment of CTD. Mechanistic understanding of how and where Cr functions in the brain is poorly documented although would be crucial to develop efficient therapies. While several rodent models of CTD have been developed, most studies focused on the description of behavioral and cognitive deficits caused by a lack of brain Cr and have not investigated the molecular underpinnings of this disorder. In addition, while these models do appear to recapitulate the phenotype of humans with CTD (*Baroncelli et al., 2014*; *Duran-Trio et al., 2021*; *Skelton et al., 2011*), the important translational gap between rodents and humans needs to be addressed. In the case of this disorder, human post-mortem samples may be difficult to obtain and frequently lack proper controls, thus limiting the investigation of molecular mechanisms. Thus, there is a great need for a relevant biological model to bridge the translational gap in our understanding of CTD. The generation of a comprehensive human model system, recapitulating some of the clinical features of CTD and providing translationally relevant data, would be a significant advancement in our potential to explore the molecular pathomechanism of CTD and validate treatment methodologies. The recent development of specific brain organoids derived from induced pluripotent stem cells (iPSCs) provides such a possibility.

In the present work, we have generated iPSCs from fibroblasts of three CTD patients, using previously published methods (*Nassor et al., 2020*; *Pavoni et al., 2018*). We examined the morphology, mRNA expression, and proteomic profile of the obtained brain organoids. Several pathways have been identified related to brain development that are altered in CTD-derived brain organoids. Transfecting a functional *SLC6A8* gene into CTD patient-derived iPSCs rescued Cr uptake and normalized much of the proteomic and other pathological features. This study establishes for the first time the use of human CTD brain organoids as a high-fidelity model for understanding the pathophysiology of this devastating disease. The use of human brain organoids from CTD patients is technically innovative and, together with the proteomic studies, provides new information on the cellular basis of cerebral Cr and the molecular mechanism underlying the onset of the CTD phenotype. Here, we also provide a database of the cell-specific alterations in CTD-related molecular pathways of translational value, relevant to treatment design.

## Results

### Generation and characterization of CTD brain organoids

To develop CTD brain organoids, we generated iPS cell lines from the fibroblasts of three CTD patients (CTD 1–4, CTD 2–3, and CTD 3–7). Control iPSCs were derived from the fibroblasts of three healthy volunteers (BJ, SP, and PK *Pavoni et al., 2018*; *Roux et al., 2019*; *Trotier-Faurion et al., 2013*). All *SLC6A8* gene mutations and the disease phenotypes were described in detail by *Valayannopoulos et al., 2013*.

We found that all iPSC lines expressed the pluripotency factors SOX2, NANOG, and OCT4, unlike fibroblasts (*Figure 1—figure supplement 1A*), as well as the pluripotency marker cell surface antigens SSEA4 and TRA1-60 (*Figure 1—figure supplement 1B*). In addition, the iPSC lines generated all three embryonic germ layers (ectoderm, endoderm, and mesoderm) in teratoma formation (*Figure 1—figure supplement 1C*). As shown in *Figure 1A–D*, basal Cr levels (0) were 4–8 times lower in CTD-derived iPSCs compared with control iPSCs. To confirm the reduced Cr uptake, we supplemented iPSCs with 25, 75, or 125 µM of Cr. Healthy iPSCs exhibited a dose-dependent Cr uptake (*Figure 1A*). However, the amount of Cr in CTD-derived iPSCs was similar at any concentration of Cr-supplemented media, further demonstrating the lack of a functional SLC6A8 (*Figure 1A*). These experiments clearly demonstrated that the CRT is necessary for Cr uptake into patient-derived iPSCs.

To examine the specific role of the CRT, we generated CTD-rescue iPSCs by transfecting a functional *SLC6A8* gene into the CTD patient-derived iPSCs. We used a transposon-based stable system for expressing *SLC6A8* along with green fluorescent protein (eGFP) for the selection and cloning of the transporter-expressing iPSCs (*Figure 1—figure supplement 2A, B* and B). The CTD-rescue iPSCs transported Cr from the media, even exceeding the uptake rate seen in the healthy control-derived iPSCs (*Figure 1B–D*).

Healthy and CTD patient-derived iPSCs were differentiated into brain organoids (*Figure 1E*). In agreement with our previous results, as well as those in the literature (*Nassor et al., 2020*; *Pavoni et al., 2018*), the iPSC-derived brain organoids showed an expression pattern consistent with a telencephalic regionalization (*Figure 1F*). When following the organoid development, there were no differences in the growth pattern or diameter of healthy and CTD brain organoids (*Figure 1G*). We assessed the intra and interproduction variability of the generated organoid models by comparing the size of the brain organoids in the different experiments (*Figure 1—figure supplement 3*) and found an intraproduction variation of 9 to 27%, and an interproduction variability of 13%.

Consistent with the iPSCs, CTD brain organoids had reduced Cr levels at basal conditions (0) and when supplemented with 75 µM Cr media (*Figure 1H, I*). We also used iPSCs from CTD patients stably expressing a functional SLC6A8 protein and derived brain organoids from these cells. SLC6A8 expressing brain organoids showed GFP fluorescence in the whole area of the organoid (*Figure 1—figure supplement 2C*). The basal Cr concentration in SLC6A8 expressing brain organoids was comparable with the organoids obtained from healthy controls. Furthermore, media Cr supplementation increased the intracellular Cr levels in these organoids, suggesting that they have a functional SLC6A8 protein (*Figure 1H, I*). These experiments demonstrated that the CTD-derived brain organoids exhibit a functional deficit in creatine uptake, and this deficiency can be overcome by the re-establishment of a functional CRT.

### CTD brain organoids show dysregulation of neurogenesis

To explore the potential cell-specific changes associated with CTD pathology, we examined the expression of several neuronal and developmental markers in the brain organoids using immunohistochemistry (IHC) and real-time PCR (RT-PCR). As shown in *Figure 2A*, IHC identified major cellular markers both in the two months old healthy-derived and CTD-derived organoids. Brain organoids expressed markers for neural progenitor cells (SOX2 and PAX6), intermediate progenitors (β-III tubulin), immature neurons (TBR1), mature neurons (MAP2 and NeuN), and astrocytes GFAP (*Figure 2A*). Distinct SOX2 and PAX6 positive progenitor cells were distributed surrounding ventricle-like structures (*Figure 2A*). In addition, brain organoids expressed the post-synaptic marker PSD95 (*Figure 2A*). Although, all markers were observed in both groups, there were noticeable differences in the morphology of the CTD organoids compared with the control. The expression of progenitor cell markers SOX2 and PAX6 in the healthy and CTD brain organoids showed that the ventricle-like structures were less organized and defined in the CTD brain organoids.

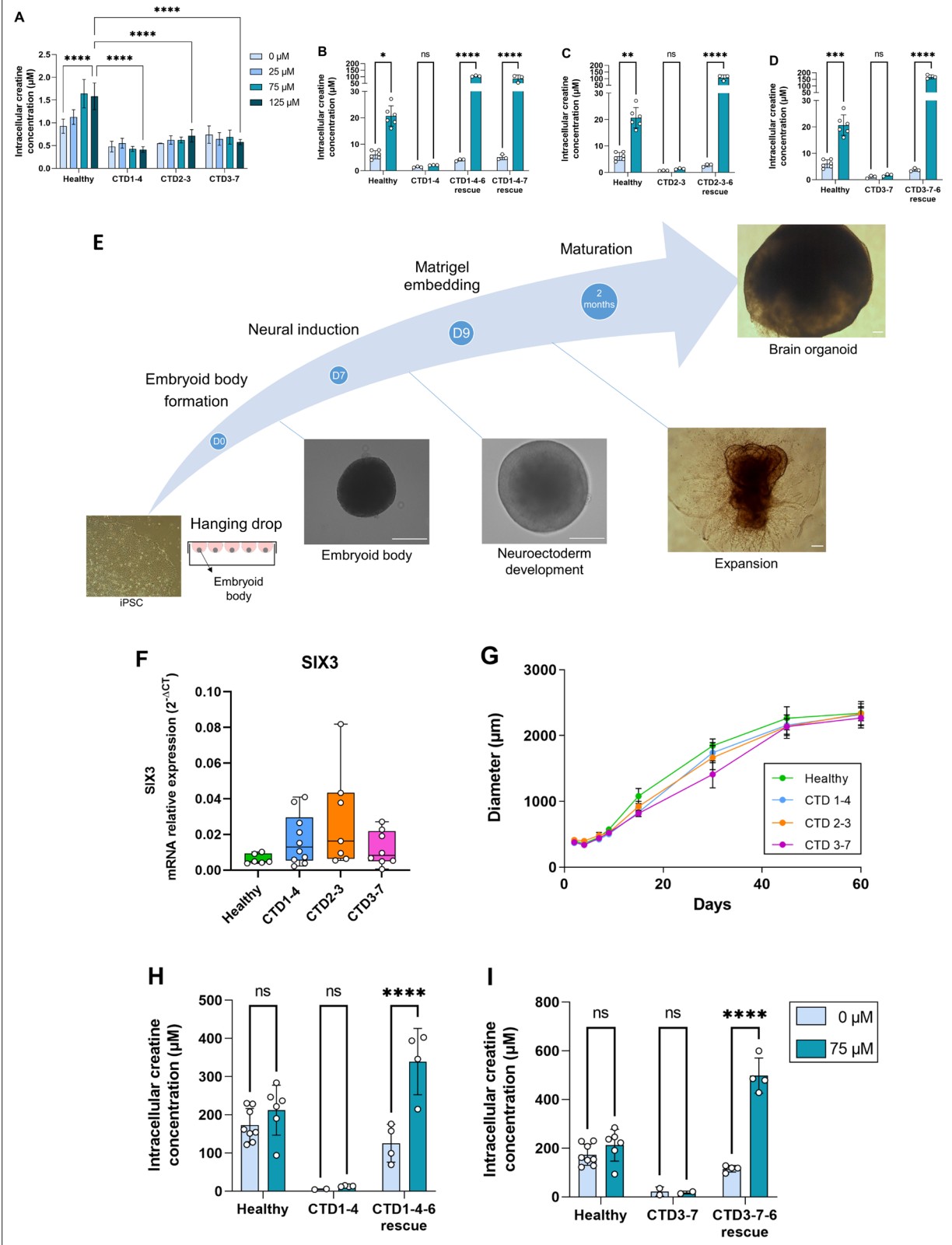

**Figure 1.** Generation and characterization of human creatine transporter deficiency (CTD) brain organoids. (**A**) Intracellular concentration of creatine in CTD iPSCs after 1 hr incubation with creatine-supplemented media. Healthy control is BJ. n=3–4, one-way ANOVA, Tukey's multiple comparison test. (**B–D**) Intracellular concentration of creatine in CTD-rescue iPSCs CTD1-4 (**B**), CTD2-3 (**C**), and CTD3-7 (**D**) after 1 hr of incubation with creatine-supplemented media. Healthy controls are BJ and SP. n=3, two-way ANOVA, Šídák's multiple comparisons test. (**E**) Schematic protocol of brain organoid

*Figure 1 continued on next page*

*Figure 1 continued*

development from iPSCs. Representative images are shown for each stage. Scale bar: 200 µm. (**F**) Relative mRNA expression of the telencephalon marker SIX3. n=6–10 (**G**) Quantification of the diameters of healthy (BJ) and pathological brain organoids at different time points. n=2–22 brain organoids per production, 4–6 productions per cell line. (**H–I**) Intracellular concentration of creatine in brain organoids and CTD-rescue brain organoids CTD1-4 (**H**) and CTD3-7 (**I**) after 6 h of incubation in creatine-supplemented media. Healthy controls are BJ and SP. n=2–4, two-way ANOVA, Šídák's multiple comparisons test.

The online version of this article includes the following figure supplement(s) for figure 1:

**Figure supplement 1.** Generation of creatine transporter deficiency (CTD) iPSCs.

**Figure supplement 2.** Generation of creatine transporter deficiency (CTD)-rescue iPSCs.

**Figure supplement 3.** Assessment of brain organoids variability.

These morphological studies were extended by examining mRNA expression levels of specific marker genes. Consistent with the morphological alterations, CTD organoids had lower mRNA expression of the neural progenitor markers SOX2 (0.0061<p<0.0408) and PAX6 (0.0001<p<0.0019), compared with the control (*Figure 2B*). To determine if this led to dysregulated differentiation of progenitor cells, we quantified the mRNA expression of GABAergic (GABBR1) and glutamatergic (GRIA2) markers. We found a significant reduction in the mRNA expression of the GABAergic [GABBR1 (p=0.0392)] and glutamatergic markers [GRIA2 (0.0041<p<0.0487), as well as of vGluT1 (0.0001<p<0.0241)], in the CTD brain organoids, compared with controls (*Figure 2B*). These findings suggest that a loss of progenitor cells (SOX2+, PAX6+) leads to dysregulation of neuronal differentiation with a reduction of GABAergic and glutamatergic neuron generation.

Next, we characterized the impact of loss of SOX2 and PAX6 progenitor cells on synapse formation by measuring mRNA expression levels of PSD95 and CREB in the brain organoids (*Figure 2B*). We found a significant down regulation of PSD95 and CREB mRNA in CTD brain organoids, as compared to the controls (0.0011<p<0.0060 and p=0.0433, respectively). Together, these results suggest that CTD brain organoids are well organized, showing the diversity of cell types but demonstrate a deficient neurogenesis. These findings are consistent with the data in a murine CTD model showing a decrease in synaptic markers associated with cognitive deficit (*Ullio-Gamboa et al., 2019*) as well as the reduced mRNA expression of SOX2 seen here (*Figure 2—figure supplement 1*).

## Proteomics and gene set arrangement (GSEA) analyses in healthy and CTD brain organoids show pathways involved in neurogenesis dysregulation

For an in-depth analysis of the proteins from CTD brain organoids associated with the deficit of neurogenesis in CTD patients, we carried out label-free shotgun proteomics measurements in samples obtained from healthy and CTD brain organoids (*Figure 3—figure supplement 1*). High-resolution tandem mass spectrometry on the 16 biological samples generated a very large dataset comprising a total of 943,656 MS/MS spectra. A total of 32,181 peptide sequences were listed, allowing us to monitor the abundance of 4219 proteins (*Supplementary file 1*). Following unsupervised filtering and normalization (*Hachim et al., 2020*; *Hamoudi et al., 2010*), significant changes in protein abundances between CTD-derived and healthy brain organoids were identified using a modification of the R package for ROTS (*Suomi et al., 2017*). A total of 2468, 2492, and 2479 proteins from CTD patient organoids 1, 2, and 3, respectively, were selected as the most confident (*Supplementary file 2*). Reproducibility plots, PCA, volcano plots, and heatmaps generated using unsupervised hierarchical clustering, assessing the degree and quality of data separation between the different groups, are presented in *Figure 3A–R*. The differential expression analysis of the normalized and filtered proteins between healthy and brain organoids from CTD patients, identified 712, 658, and 597 proteins as significantly altered in brain organoids from patients 1, 2, and 3, respectively (p-*value* <0.05 and FDR ≤0.25) (*Supplementary file 3*). A total of 510 proteins were altered in all three CTD patients compared with the healthy brain organoids, while 116, 52, and 21 proteins were found to be specifically altered only in patient 1, 2, and 3, respectively (*Figure 3S–U*).

To identify the enriched pathways altered in CTD brain organoids, absolute GSEA was performed on the following three pairs of conditions: Healthy (BJ) vs brain organoids from CTD patient (CTD1_4); Healthy (BJ) vs brain organoids from CTD patient (CTD2_3); Healthy (BJ) vs iPSC brain organoids from

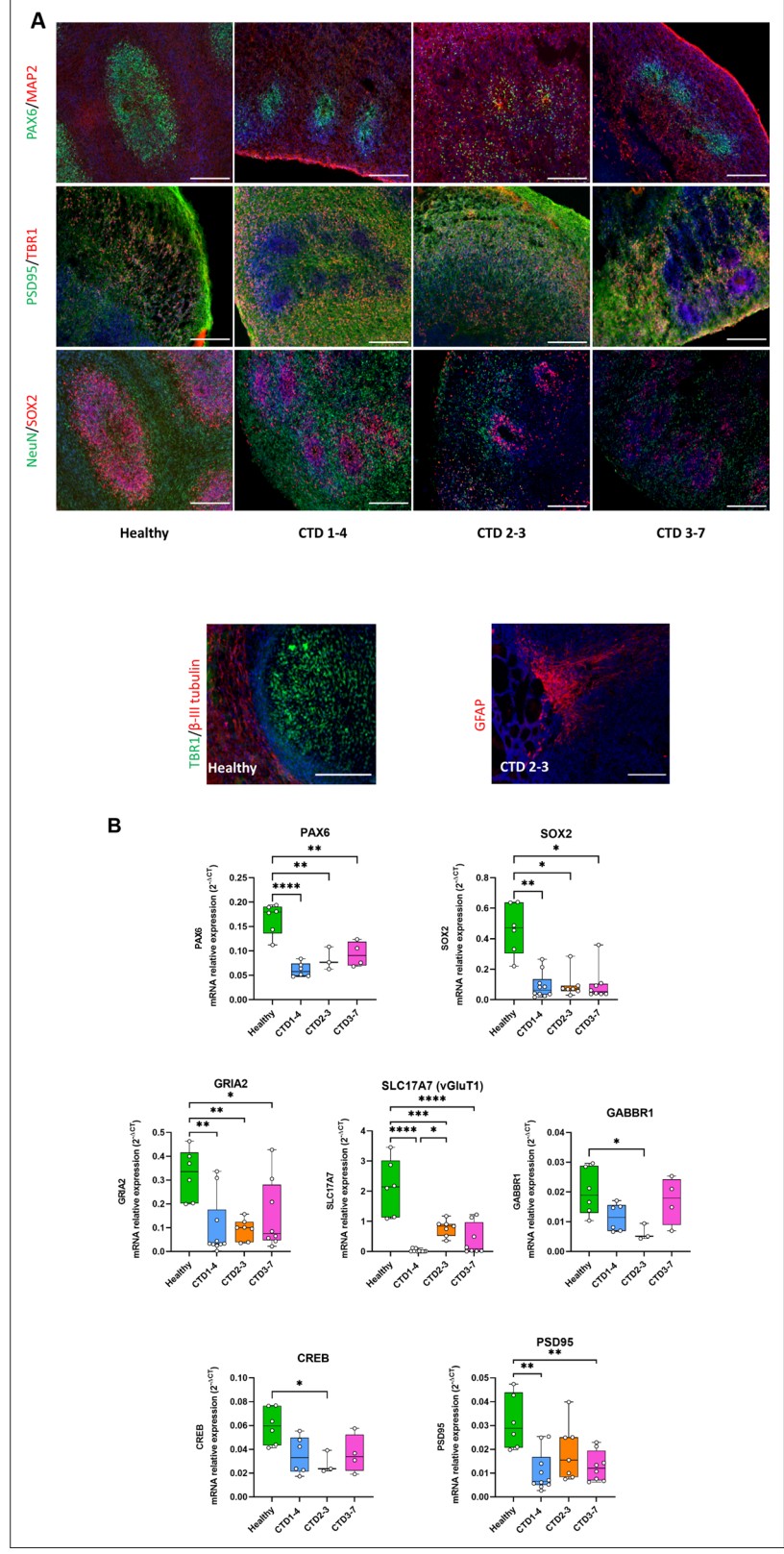

**Figure 2.** Creatine transporter deficiency (CTD) brain organoid organization and neurogenesis deficit.
(**A**) Representative images of 2 month old brain organoid tissue sections immunostained for PAX6 (radial glial cell and forebrain marker), MAP2 (neuronal marker), PSD95 (post-synaptic marker), TBR1 (immature neuron marker), NeuN (neuronal marker), SOX2 (radial glial cell marker), β-III tubulin (intermediate progenitor marker), GFAP

*Figure 2 continued on next page*

*Figure 2 continued*

(astrocyte marker). DAPI marks nuclei in blue. Scale bar: 200 μm (**B**) Relative mRNA expression of PAX6 and SOX2 (radial glial cell markers), GRIA2 and vGluT1 (glutamatergic markers), GABBR1 (GABAergic marker), PSD95 (post-synaptic marker) and CREB. n=3–6, one-way ANOVA, Tukey's multiple comparison test.

The online version of this article includes the following figure supplement(s) for figure 2:

**Figure supplement 1.** Neurogenesis deficit in a creatine transporter deficiency (CTD) mouse model.

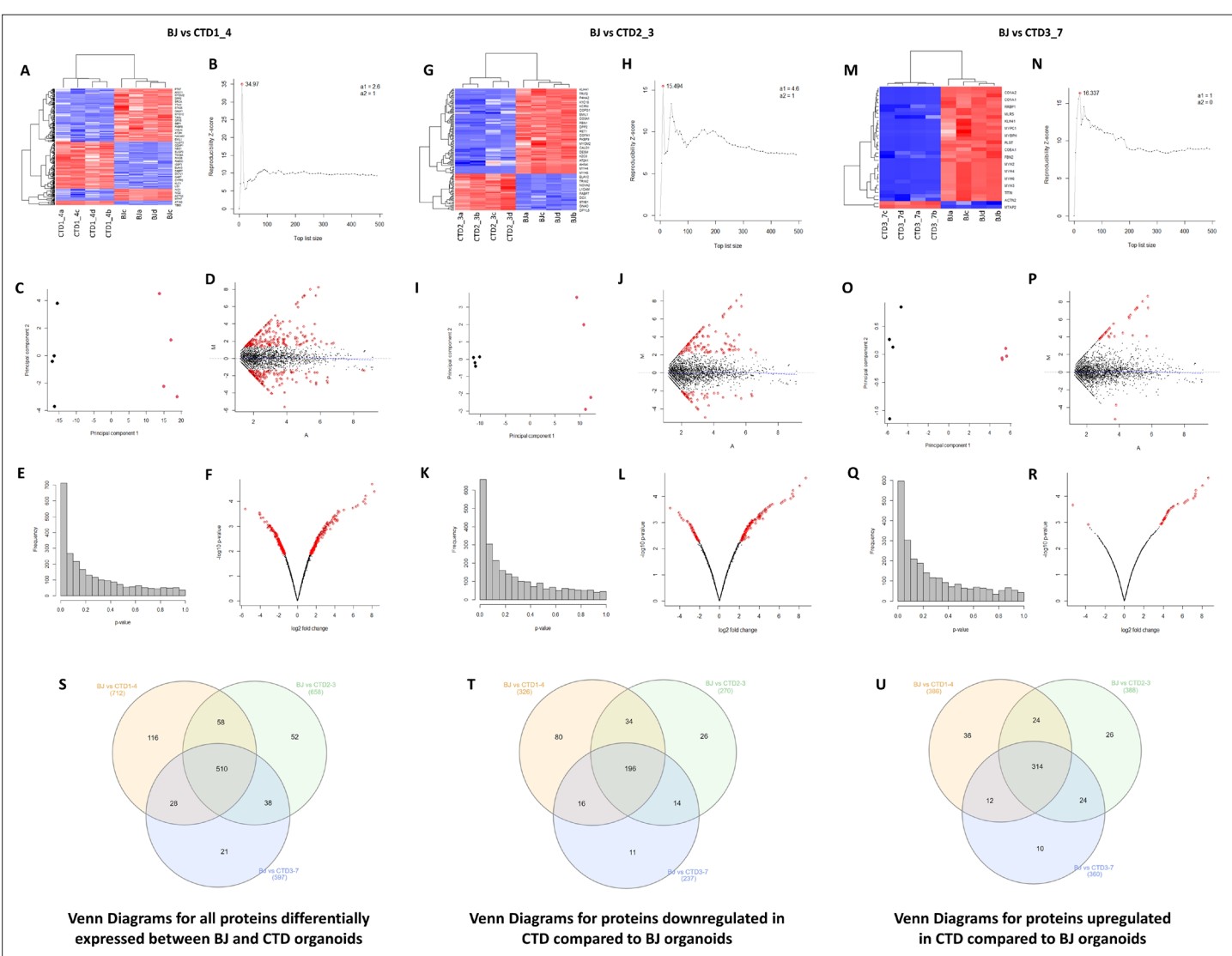

**Figure 3.** Unsupervised hierarchical clustering, reproducibility plots, principal component analysis (PCA), volcano plots, and Venn diagram showing overlap of the proteins. (**A–R**) The degree and quality of the separation of the data between the various groups being compared Healthy (BJ) vs creatine transporter deficiency (CTD)-derived brain organoids (CTD1_4); Healthy (BJ) vs CTD-derived brain organoids (CTD2_3); Healthy (BJ) vs CTD-derived brain organoids (CTD3_7) was assessed using unsupervised hierarchical clustering, reproducibility plots and PCA, and the differentially expressed proteins were visualized using volcano plots. (**S–U**) Venn diagram showing overlap of the differentially expressed proteins between healthy and CTD-derived brain organoids identified using a modification of the R package for ROTS. (**S**) Venn diagrams for all proteins differentially expressed between BJ and CTD organoids. (**T**) Venn diagrams for proteins downregulated in CTD compared to BJ organoids. (**U**) Venn diagrams for proteins upregulated in CTD compared to BJ organoids.

The online version of this article includes the following figure supplement(s) for figure 3:

**Figure supplement 1.** Flowchart diagram depicting the entire bioinformatics analysis workflow.

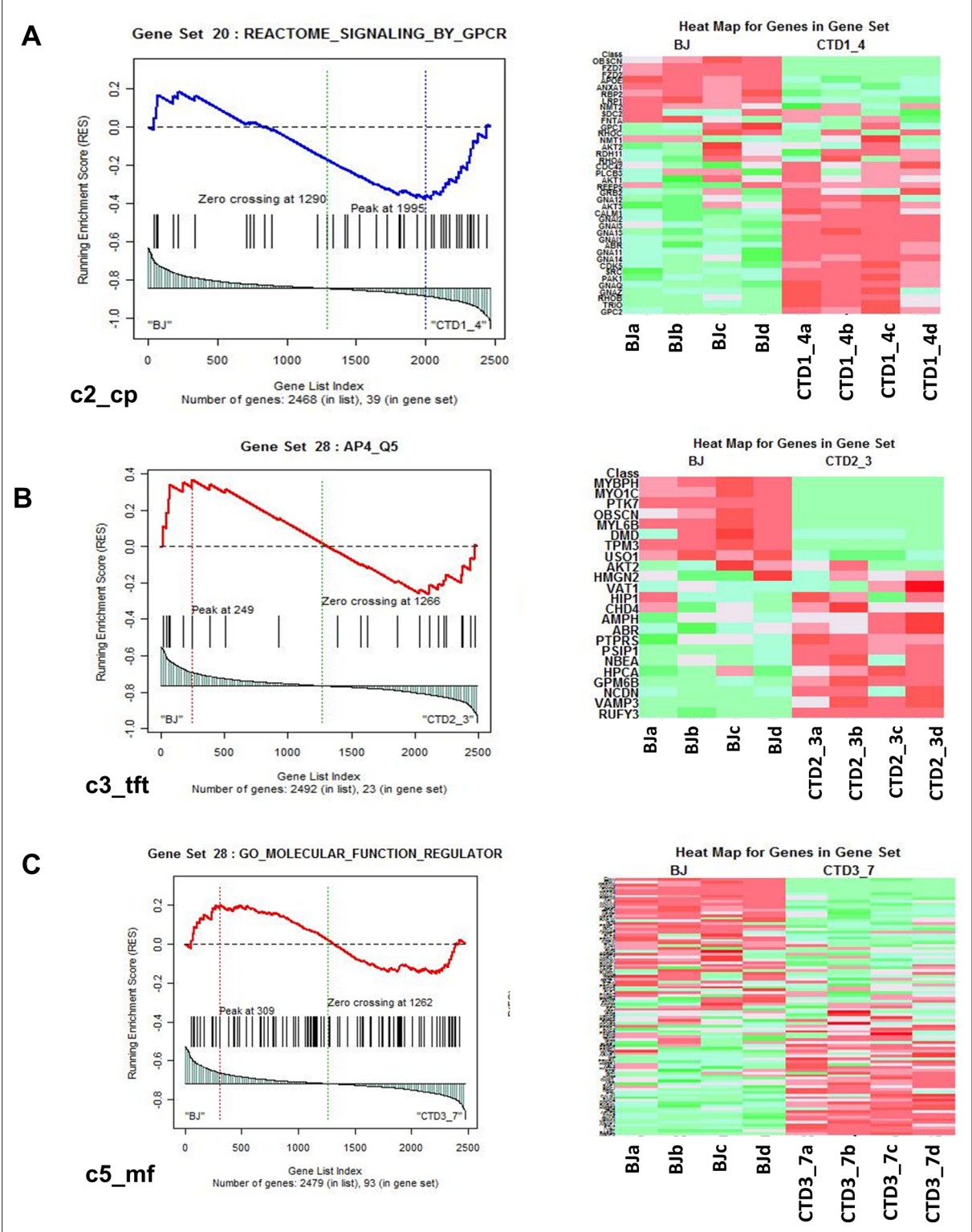

**Figure 4.** Representation of heatmaps and graphs obtained by the absolute gene set enrichment analysis (GSEA) for significant pathways with enrichment scores. (**A–C**) Enrichment score and graphical representation for the GSEA for healthy vs creatine transporter deficiency (CTD)-derived brain organoids obtained across the gene sets c2_cp; c3_tft; c4_cgn; c5_bp; c5_mf; c7.

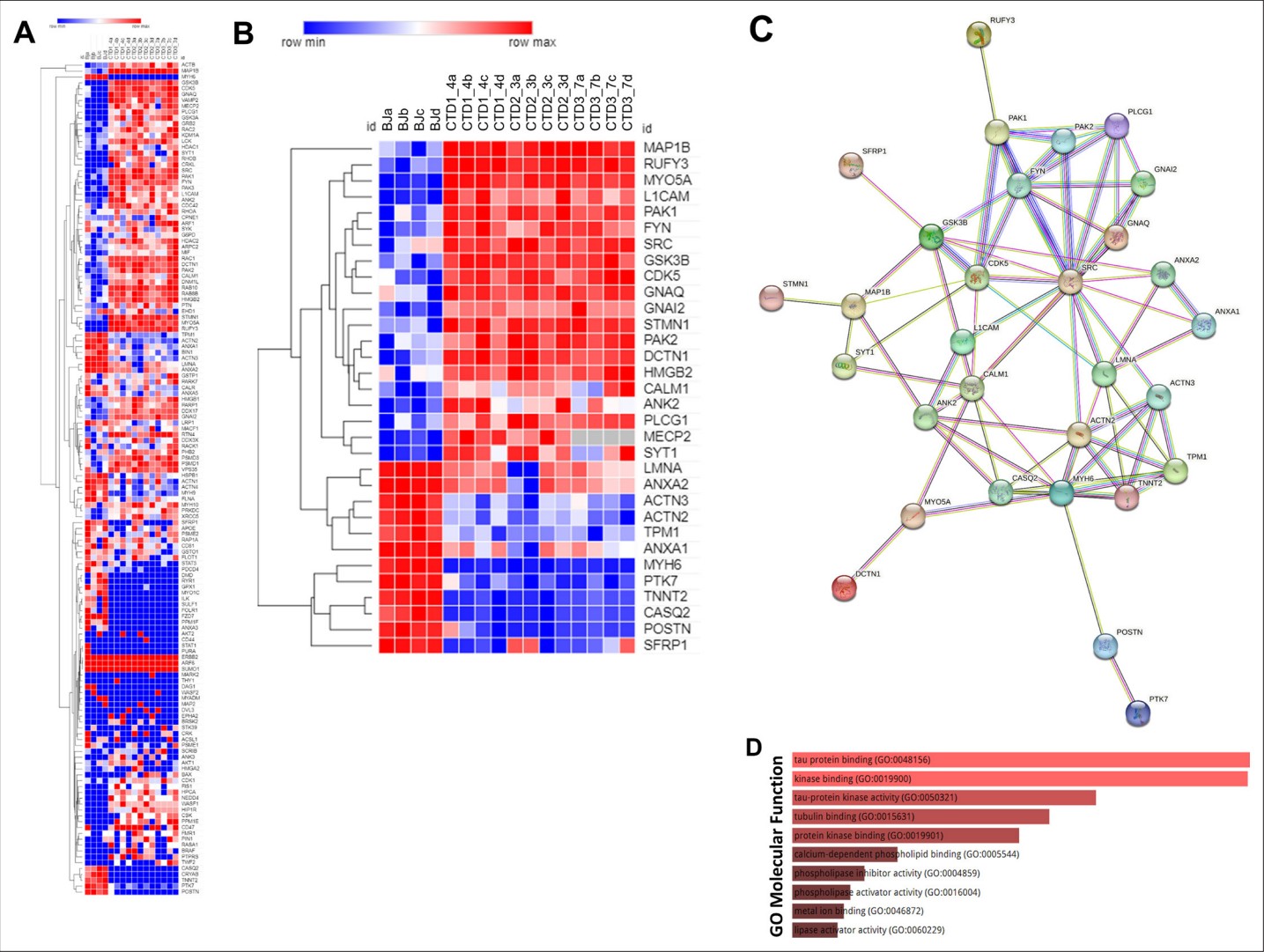

**Figure 5.** Heatmap, STRING interaction, and GO molecular functional analysis. (**A**) Heatmap showing expression levels of the 142 proteins in the top 90 percentile. (**B**) Heatmap showing expression levels of 32 most altered and abundant proteins were found to be significantly altered in CTD-derived cerebral organoids compared to normal organoids. (**C**) STRING interaction of the 30 proteins selected. (**D**) GO Molecular Functional analysis obtained using Enrichr on the 32 proteins.

CTD patient (CTD3_7). The analysis identified that 22, 199, 10, 388, and 323 pathways derived from $c2$, $c3$, $c4$, $c5$, and $c7$, respectively, are significantly enriched in BJ vs CTD1_4; 32, 199, 8, 416, and 498 pathways derived from $c2$, $c3$, $c4$, $c5$, and $c7$ are significantly enriched in BJ vs CTD2_3; and 26, 182, 8, 474, and 541 pathways derived from $c2$, $c3$, $c4$, $c5$, and $c7$ are significantly enriched in BJ vs CTD3_7. An example of a representation of the output from the GSEA for each gene set is shown in *Figure 4*. For each significant pathway, enriched proteins were identified and their recurrence or frequency in other pathways among all studied proteins was searched. Protein frequency can be defined as the number of times a protein occurs across all the enriched components from the significantly enriched pathways. The proteins with the highest frequency across the multiple significant pathways using the 90 percentile as cut-off was selected. This frequency analysis identified 142 differentially abundant proteins occurring frequently across all enriched pathways (*Supplementary file 4*).

## Stepwise regression statistical modeling identified GSK3β and SRC cascade signaling associated with CTD brain organoids

GSEA and frequency analysis (90 percentile cut-off) identified 142 differentially abundant proteins occurring frequently across all enriched pathways (*Supplementary file 4*). A heatmap showing the

abundance levels of the 142 proteins in the top 90 percentile is represented in *Figure 5A*. To further reduce the set of available proteins, the 142 proteins identified from the GSEA and the frequency analysis were sorted according to fold change; the most abundant proteins with fold change (–3<fold change >+3) (48 proteins) and detected with at least 10 MS/MS spectra (32 proteins) were retained (*Supplementary file 5*). A heatmap showing the relative abundance of the 32 selected proteins is represented in *Figure 5B*. Among the 32 proteins highlighted, 20 proteins are upregulated in CTD vs healthy brain organoids, and 12 proteins are downregulated.

An ENRICHR analysis was then performed on these selected proteins to identify those potentially related to the clinical features of CTD patients (*Supplementary file 6*). The ENRICHR analysis on the 32 selected proteins revealed that the GSK3β and SRC cascade signaling has the highest frequency (*Supplementary file 6*). Noteworthily, 21 proteins are related to autism spectrum disorders (e.g. PAK2), intellectual disability (e.g. PAK1), neurodevelopmental disorders (e.g. GSK3β) and epileptic encephalopathies (e.g. MAP1B) (*Supplementary file 7*).

## The interplay between GSK3β cascade and neurogenesis deficit in CTD brain organoids

The Search Tool for the Retrieval of Interacting Genes (STRING) webtool was used to further examine the significant enrichments of protein-protein interactions and establish pathways involved in CTD

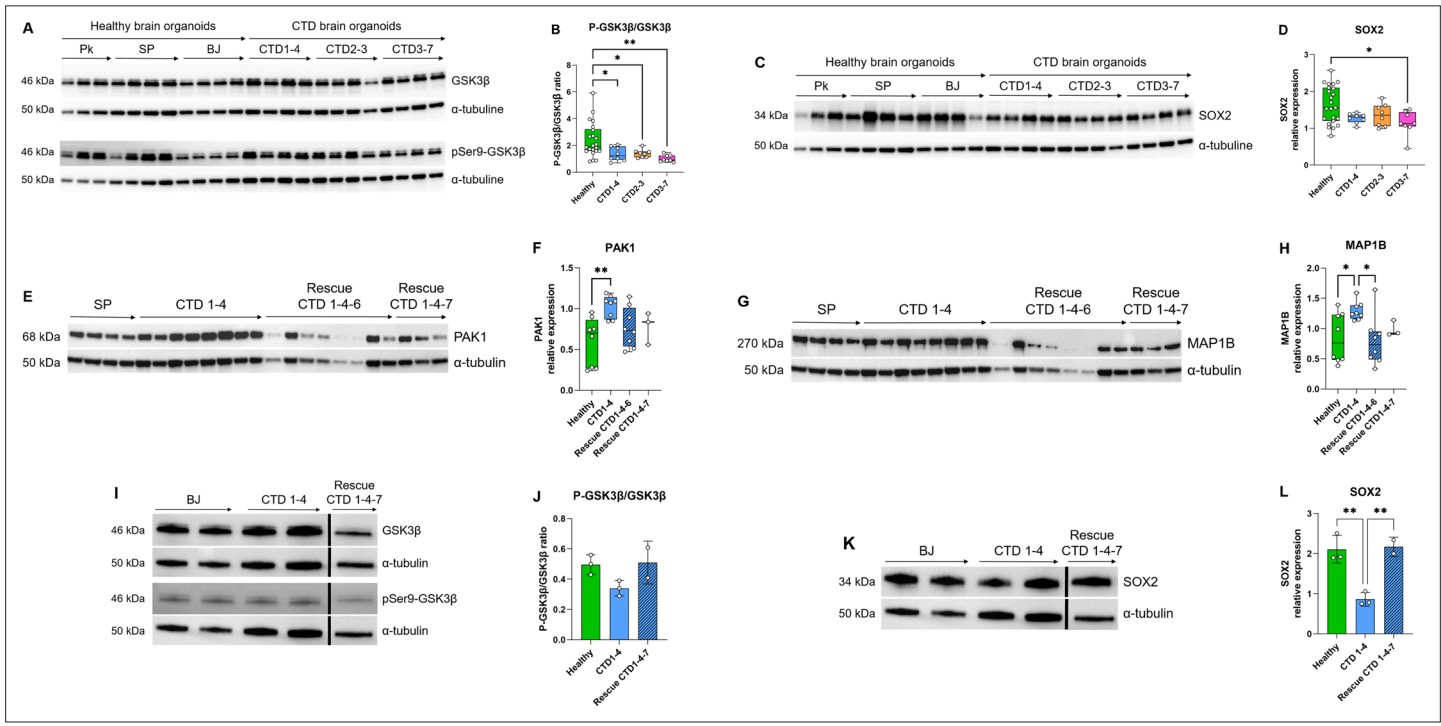

**Figure 6.** Relationship between GSK3β and neurogenesis deficit. (**A–B**) Representative western blot of GSK3β and pSer9-GSK3β in creatine transporter deficiency (CTD) vs healthy brain organoids (**A**) and graph showing analysis of P-GSK3β/GSK3β ratio (**B**). n=8, one-way ANOVA, Dunnett's multiple comparison test. (**C–D**) Representative western blot of SOX2 in CTD vs healthy brain organoids (**C**) and graph showing analysis of SOX2 relative expression (**D**). n=8, one-way ANOVA, Tukey's multiple comparison test. (**E–L**) Representative western blot of PAK1 (**E**), MAP1B (**G**), GSK3β and pSer9-GSK3β (**I**), and SOX2 (**K**) in brain organoids obtained from CTD iPSCs and CTD-rescue iPSCs (CTD 1–4), and graphs showing analysis of PAK1 (**F**), MAP1B (**H**), P-GSK3β/GSK3β ratio (**J**), SOX2 (**L**) relative expression. n=2–8, one-way ANOVA, Dunnett's multiple comparison test. (**I and J**) the lanes were run on the same gel but were noncontiguous.

The online version of this article includes the following source data for figure 6:

**Source data 1.** Original files of the full raw unedited blots (colorimetric and chemiluminescence images) for *Figure 6A–B*.

**Source data 2.** Original files of the full raw unedited blots (colorimetric and chemiluminescence images) for *Figure 6C–D*.

**Source data 3.** Original files of the full raw unedited blots (colorimetric and chemiluminescence images) for *Figure 6E–H*.

**Source data 4.** Original files of the full raw unedited blots (colorimetric and chemiluminescence images) for *Figure 6I–L*.

**Source data 5.** Figures with the uncropped blots with the relevant bands labeled for *Figure 6*.

pathophysiology. The connections between the top proteins involved in the clinical features of CTD patients are shown in *Figure 5C*. We focused on GSK3β pathway because this key kinase is implicated in many cellular processes such as neurogenesis (*Hur and Zhou, 2010*; *Kim et al., 2009*). Ser9 phosphorylation in GSK3β is known to inhibit the kinase activity. We systematically analyzed the level of inactive kinase by immunoblot with a specific antibody. There was a significant reduction in Ser9-phosphorylation in GSK3β in CTD brain organoids compared with healthy brain organoids (0.0034<p<0.0426) (*Figure 6A and B*) as well as a significant reduction in SOX2 abundance in the same CTD brain organoids (p=0.0146) (*Figure 6C and D*). Moreover, we confirmed the increased abundance of two proteins highlighted by proteomics analysis, PAK1 (p=0.0034) and MAP1B (p=0.0331), with an additional healthy cell line (SP) and new productions of CTD brain organoids (*Figure 6E–H*). PAK1 and MAP1B abundance (p=0.0173) were decreased in the *SLC6A8*-expressing CTD-derived organoids (*Figure 6E–H*), while Ser9-phosphorylation in GSK3β and SOX2 abundance (0.0003<p<0.0072) were increased (*Figure 6I–L*). This suggests that these changes are mediated through the loss of the functional transporter and pointed out the major role of Cr in the regulation of cerebral proteins in CTD brain organoids.

## Discussion

The aim of this study was to establish a new translational model for CTD to better characterize the pathophysiology of this disease. We generated brain organoids derived from iPSCs of CTD patients. As predicted, the iPSCs and brain organoids from CTD patients were not able to transport Cr into the cells from the media. The iPSC-derived CTD brain organoids showed similar size and cell type diversity to those obtained from healthy donors, however, they showed decreased expression of neurogenesis-related markers. Normal brain organoids are composed of mature and developing neurons, as well as astrocytes (*Lancaster et al., 2013*; *Nassor et al., 2020*; *Pavoni et al., 2018*). The organoids derived from CTD patients show a similar composition with neural progenitor cells located in ventricle-like structures marked by SOX2 and PAX6, and immature and mature neurons marked by TBR1 and NeuN that have migrated from the ventricles. The transcription factors (SOX2 and PAX6) are critical regulators of neuronal stem cell differentiation and proliferation, ensuring the successful process of neurogenesis (*Cimadamore et al., 2013*; *Manuel et al., 2015*). SOX2 mutations are associated with intellectual disabilities (*Dennert et al., 2017*), seizures, and defective hippocampal development (*Mercurio et al., 2021*).

In addition to the decreased expression of the progenitor markers, we show that CTD-derived organoids have reduced expression of GABAergic (GABBR1), glutamatergic (GRIA2), and postsynaptic (PSD95) markers, suggesting a reduction in synaptogenesis. This is in agreement with our findings in *Slc6a8* knockout (*Slc6a8$^{-/y}$*) mice that showed reduced mRNA expression of PSD95 and CREB (*Ullio-Gamboa et al., 2019*). Reduced cortical spine density and reductions in protein levels of several synaptic markers have been observed in the brains of *Slc6a8$^{-/y}$* mice and rats (*Chen et al., 2021*; *Duran-Trio et al., 2022*). Thus, our CTD brain organoid model recapitulates the altered neurogenesis seen in murine CTD models and this impaired neurogenesis may lead to the cognitive deficits in CTD. Udobi et al., showed that developmental deletion of the *Slc6a8* gene in mice is necessary to gain the same cognitive deficits seen in the constitutive *Slc6a8$^{-/y}$* mice, providing further evidence for the importance of Cr in brain development (*Udobi et al., 2019*). Our findings are in agreement with previous observations in Down syndrome, showing the defect in neurogenesis and consequently the diminished proliferation and decreased expression of layer II and IV markers in cortical neurons in the subcortical regions (*Tang et al., 2021*).

We performed proteomic analysis to identify proteins and molecular pathways that may be disrupted in CTD. Organoids from CTD patients showed altered expression of several proteins compared with normal brain organoids. We highlighted 32 proteins with altered abundance in comparison with healthy brain organoids. Among these 32 proteins, 21 proteins have already been related to autism spectrum disorders, intellectual disability, neurodevelopmental disorders, and epileptic encephalopathies (*Supplementary file 7*). These altered, relevant protein expression patterns indicate that CTD brain organoids may appropriately reflect the molecular physiopathology of the patients. We used a stepwise regression statistical model and ENRICHR analysis to reveal that two key pathways, GSK3β and SRC, were altered in CTD brain organoids. The search tool for the retrieval of interacting genes

(STRING) web tool confirmed that the two pathways were associated with the most abundant proteins involved in neurodevelopmental disorders.

Next, we performed a functional study on the relationship between GSK3β and neurogenesis deficit in CTD organoids. GSK3β is a constitutively active key kinase, negatively regulated by phosphorylation at Ser9 (*Hur and Zhou, 2010*). GSK3, having a broad range of substrates, regulates a wide spectrum of cellular processes, including neurogenesis, neuronal polarization, and axon growth during brain development (*Hur and Zhou, 2010*). This regulation occurs either by direct action or through transcriptional modulations. As an example, active GSK3β directly modulates microtubule dynamics by phosphorylating microtubule-associated proteins (MAPs) (*Rippin and Eldar-Finkelman, 2021*), such as MAP1B which was highlighted in our proteomic analysis. MAP1B in turn participates in the regulation of the structure and physiology of dendritic spines in glutamatergic synapses (*Tortosa et al., 2011*). GSK3β can also modulate gene expression by controlling the level, nuclear localization, and the DNA binding of transcription factors, thus indirectly regulating neurogenesis (*Hur and Zhou, 2010*). As an example, the cAMP response element-binding protein (CREB) is a substrate of GSK3 (*Hur and Zhou, 2010*). Thus, the overactivation of GSK3β in CTD brain organoids is consistent with our observations in a CTD mouse model, where we showed that CREB was downregulated at the transcriptional level (*Ullio-Gamboa et al., 2019*). Numerous studies have shown that modulation of GSK3β activity can regulate neural progenitor homeostasis (*Dohare et al., 2019*; *Guyot et al., 2020*; *Hur and Zhou, 2010*; *Jurado-Arjona et al., 2016*; *Kim et al., 2009*) and GSK3 is also implicated in energy homeostasis, apoptosis and autophagy, as well as in interactions with neurodegeneration-linked proteins (*Hernandez et al., 2013*).

The overactivation of GSK3β in our CTD brain organoids was associated with a decreased level of neural progenitor and synaptic markers as compared to the normal organoids. We document here that this phenotype is rescued by the re-establishment of a functional SLC6A8 transporter, restoring Cr uptake in the CTD brain organoids. The introduction of a functional Cr transporter in the CTD-rescue

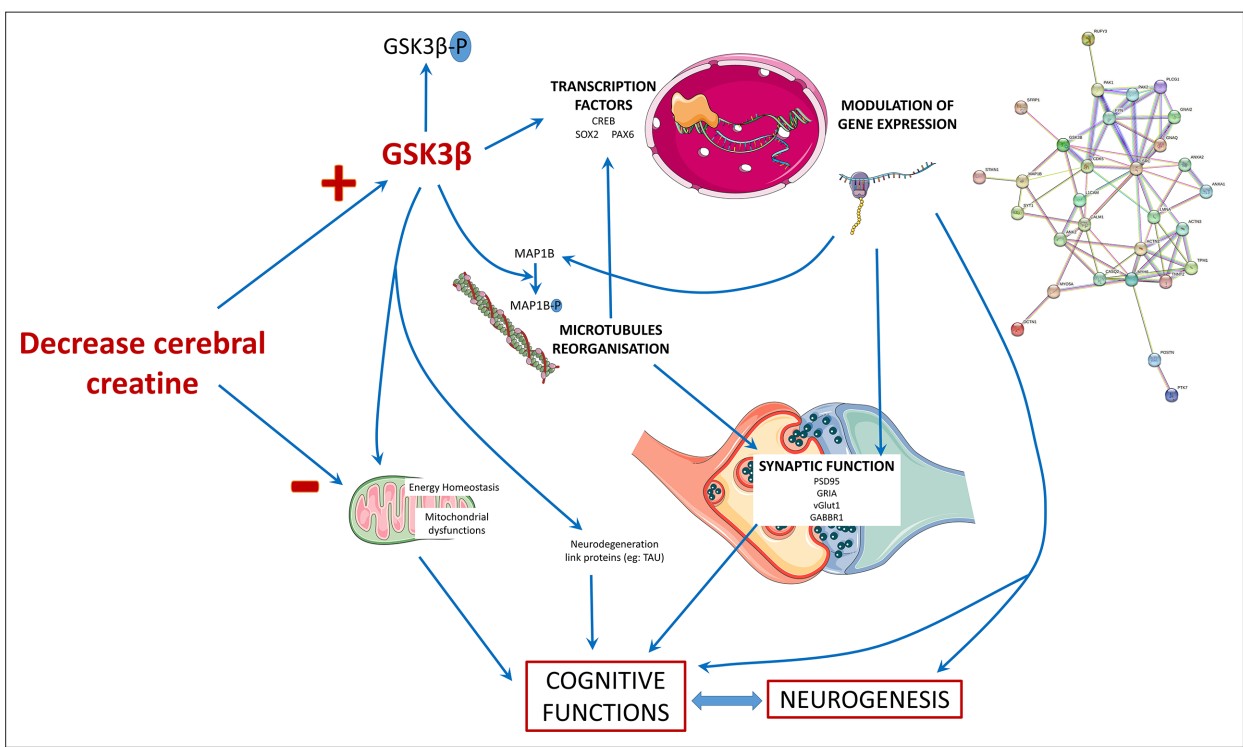

**Figure 7.** Schematic presentation of the main findings of the study. The decrease in the cerebral creatine pool could favor the accumulation of the dephosphorylated form of GSK3β, making it more active. This kinase has many targets in the cell, including transcription factors whose activity it can modulate. This modulation will in turn modify the expression of genes, which can influence several cellular processes: the reorganization of microtubules with the regulation of MAP1B, synaptic function, and neurogenesis thus affecting the functionality of brain cells and consequently cognitive functions. The alteration of neurons could also be explained by a mitochondrial dysfunction due to the decrease of creatine levels and the modulation of GSK3β activity, which is known to regulate mitochondrial activity.

organoids leads to a reduction of GSK3β activation, a restoration of the SOX2 progenitor marker level, as well as of the highlighted proteins linked to intellectual disabilities such as PAK1 and MAP1B. Alterations in GSK3 activity have been associated with many neurodegenerative diseases such as Alzheimer's disease and neurodevelopmental diseases such as autism spectrum disorders (*Hernandez et al., 2013*; *Rizk et al., 2021*). Although, the link between creatine level and GSK3β activation needs to be further examined, modulation of GSK3β activity may represent an interesting target for the treatment of this devastating intellectual disability disease.

In summary, our iPSC-derived brain organoid model recapitulates the key features of the neurodevelopmental deficits observed in CTD patients. We examined morphological and mRNA expression alterations potentially connected to the pathology of CTD. The extensive proteomics analysis pointed out 32 proteins linked to the GSK3β and SRC cascades and 21 proteins potentially associated with the clinical features of CTD patients. Using targeted molecular methodologies, we demonstrated that the altered GSK3β pathway most likely has a significant role in the mechanisms leading to a defective neurogenesis in the CTD brain organoids. The main findings of our study in CTD brain organoids are schematically summarized in *Figure 7*. We suggest that the human cellular organoid model presented here helps to understand the pathogenesis of CTD, and particularly the impact of the decreased creatine uptake into the brain on the regulation of proteins associated with the clinical features of CTD patients. Analysis of protein expression expand the knowledge about the consequences of CRT deletion and the effects of Cr replenishment, revealing new information on the molecular mechanism underlying the onset of CTD phenotype. We provide a database of the cell-specific alterations in molecular pathways of translational value relevant for treatment design. We believe that this new model of CTD patient-derived brain organoids represents an important translatable tool to examine potential treatment modalities. We generated CTD brain organoids from three families. As all patients' mutations in this study are a deletion of one amino acid, it could be interesting to generate CTD brain organoids from cells with other types of mutations to evaluate their impact on the CTD pathophysiology.

# Materials and methods
## Derivation of patient's fibroblasts
BJ primary fibroblasts were obtained from ATCC (CRL-2522). SP fibroblasts were isolated from normal placenta according to the protocol study approved by the local ethics committee (Advisory Committee for the Protection of Persons in Biomedical Research Cochin Hospital, Paris, n°18–05). Human fibroblasts from CTD patients, were obtained from skin biopsy specimens and were a gift from the Centre de Référence des Maladies Héréditaires du Métabolisme at the Necker Hospital in Paris. Three patients with cerebral creatine deficiency caused by lack of creatine transporter were studied. Informed and a written consent was obtained from all anonymized human CTD subjects, and experiments were carried out in accordance with relevant guidelines and regulations. All the mutations were previously described by *Valayannopoulos et al., 2013*:

- Patient 1: c1006_1008delAAC (pAsn336del): 9 years old showing mild mental retardation, speech delay, learning difficulties, mild behavioral impairment, and facial hypotonia.
- Patient 2: c1497_1500delGAG (pGly499del): 10 years old showing mild psychomotor retardation, speech delay, seizures, impulsivity, facial, and trunk hypotonia.
- Patient 3: c1221_1223delTTC (pGly414del): 3 years old showing no speech, autistic behavior with hyperactivity and emotional instability.

After 1–2 weeks, fibroblasts outgrowths from the explants were passaged and maintained as described previously *Trotier-Faurion et al., 2015* in Dulbecco's Modified Eagle Medium (DMEM, Life Technologies) supplemented with 10% inactivated foetal bovine serum (FBS, Life Technologies), 1% penicillin/streptomycin/neomycin (PSN, Life Technologies), 1% sodium pyruvate (Sigma-Aldrich), and 1% L-glutamine (Life Technologies).

## iPSC generation and characterization
### Reprogramming
Fibroblasts were reprogrammed using the Sendai virus reprogramming method as recommended by the manufacturer (Life Technologies). Briefly, fibroblasts were infected using SeV vectors encoding

OCT3/4, SOX2, KLF4, and c-MYC on day 0. Two days later, cells were trypsinized and picked up onto two 10 cm gelatin-coated culture dishes that had been seeded with irradiated mouse embryonic fibroblasts (irrMEFs, GlobalStem). The cultures were maintained in human embryonic stem cell (hESC) medium containing DMEM/F12 (Gibco), 20% KnockOut Serum Replacement (Gibco), 1% minimum essential medium (MEM) nonessential amino acid (Gibco), 1 mM l-glutamine (Gibco), 0.1 mM β-mercaptoethanol (Sigma-Aldrich), and 10 ng/mL of basic fibroblast growth factor (bFGF, STEMCELL technologies). Clones were picked starting on 20 days post infection and expanded on irrMEFs before being adapted to feeder-free conditions.

## hiPSC culturing

All iPSCs cell lines were maintained on hESC-qualified Matrigel (Corning) in mTeSR1 medium (STEMCELL Technologies) and passaged using mechanical dissociation at the beginning of the establishment of the cell line. Then cells were passaged using ReLeSR (STEMCELL Technologies). iPSCs were negative for mycoplasma contamination (MycoAlert, Lonza).

## Teratoma formation

To examine the developmental potential of reprogrammed clones in vivo, iPSCs grown on feeder free system were collected by dispase treatment and injected into hind limb muscle of 8-week-old immunodeficient NOD-scid IL2rynull (NSG) mice (approximately 1×6 well plate at 70% confluence for each injection). After 5 to 10 weeks, teratomas were dissected and fixed in 4% paraformaldehyde. Samples were embedded in paraffin and processed with hematoxylin and eosin staining. All studies were done with compliance with animal welfare regulations.

## Assessment of pluripotency

The pluripotency characteristics of iPSC was demonstrated (i) by assessing the presence of pluripotency marker (OCT4, SOX2, NANOG) by RT-PCR (see methods below) and (ii) by measuring the expression of pluripotency surface antigen (SSEA4 1:20, Invitrogen, 12-8843-42, and TRA1-60 1:20, Invitrogen, 12-8863-82) by flow cytometry. iPSCs were harvested by accumax treatment for 5 min until the cells detach. Cells were then fixed in paraformaldehyde 4% for 8 min. After two PBS1X washes, cells were blocked with 10% of goat serum in PBS1X (10%PBSG) for 20 min at room temperature. Cells were then incubated overnight at 4 °C with primary antibodies diluted in 10%PBSG. IgG controls (1:20, Invitrogen, 12-4742-42) were used at the same concentration. Cells were then washed twice with PBS1X-5%SVF and resuspended in FACS Flow for analysis on a BD FACSCalibur flow cytometer.

## Generation of iPS cells stably expressing SLC6A8

The full length *SLC6A8* cDNA was purchased from Addgene (pDONR221_SLC6A8, #131915) and cloned into an eGFP containing Sleeping Beauty transposon vector (p10-CAG-SLC6A8-IRES2-eGFP), where the SLC6A8 and eGFP were connected by an IRES2 sequence, providing simultaneous, but independent expression of the two proteins.

The p10-CAG-SLC6A8-IRES2-eGFP vector was electroporated together with the SB 100 X transposase (*Mátés et al., 2009*) into control and disease related iPSC lines in Amaxa 4D-Nucleofector using P4 Primary Cell 4D-Nucleofector X Kit (Lonza) with the CM133 program. After 7 days of transfection each clone was sorted based on the GFP fluorescence using BD FACSAria Cell Sorter then plated onto Matrigel-coated plates (Corning), in mTeSR1 (Stem Cell Technologies) medium containing 10 μM Y27632-2HCl (Selleckchem). Stabilized SLC6A8 and eGFP expressing clones were expanded and screened for GFP expression and for pluripotency by measuring SSEA4-APC expression by flow cytometry. The cells were labeled with anti-human SSEA-4 APC-conjugated antibody (1:100, R&D Systems) at 37 °C for 30 min. Propidium iodide (ThermoFisher Scientific) staining was employed for gating out the positively labelled dead cells. A control measurement with isotype-matched control (1:100, R&D Systems) was included. For selecting clones of SLC6A8 and eGFP expressing iPSC lines, single cell suspensions were prepared using Accutase and then the GFP expressing single cells were plated in Matrigel-coated 96-well plates in mTeSR1 medium containing 10 μM Y27632-2HCl, using BD FACSAria Cell Sorter. The clones were expanded in mTeSR1 medium. Surviving clones used for further experiments were selected based on pluripotent morphology and bright GFP fluorescence.

## Brain organoid generation and culture

We followed the protocol published by Lancaster & Knoblich with minor modifications from *Lancaster and Knoblich, 2014*; *Nassor et al., 2020*; *Pavoni et al., 2018*. At day 0, iPSCs were dissociated and counted. Hanging drops of 20 µl of embryoid body medium containing 15,000 cells per drop were cultured on a Petri dish cover to allow them to aggregate before being collected and placed in a well of a non-treated 24-well plate (Sarstedt). Media changes, neural induction and neural differentiation were performed following the published protocol (*Lancaster and Knoblich, 2014*; *Nassor et al., 2020*; *Pavoni et al., 2018*) with the use of an organoid embedding sheet for the matrigel embedding step (STEMCELL Technologies). Brain organoids were used at 2 months-old.

## Creatine uptake and quantification

iPSCs were passaged using accumax (Sigma) and seeded at 300,000 cells/well in a six-well plate. When iPSC reached confluence, they were incubated with or without Cr-supplemented medium for 1 hr, then harvested using accumax. Cr levels were quantified using Cr quantification kit (Sigma, MAK079), either the colorimetric or fluorometric version of the kit. For the colorimetric test, absorbance was measured at 570 nm with a BioTek Epoch microplate spectrophotometer (Agilent Technologies). For the fluorometric test, fluorescence intensity ($\lambda$ ex = 535 nm, $\lambda$ em = 587 nm) was measured with a SpectraMax M5 Multimode Plate Reader (Molecular Devices). Brain organoids were incubated with or without a Cr-supplemented medium for 6 hr. Cr levels were quantified using the same method as for iPSCs, with the exception of lysis for which Precellys Evolution tissue homogenizer (Bertin) was used.

Cr levels in samples were normalized by total protein quantification (Bradford assay).

## Immunofluorescence

Brain organoids were fixed in 4% paraformaldehyde for 45 min. After two washes in PBS, organoids were moved to 30% sucrose solution overnight at 4 °C. Organoids were then embedded in optimal cutting temperature (OCT) compound and flash frozen at –80 °C. Sections (20 µm) were cut with a cryostat (HM560 Microm Microtech). After permeabilization with 0.3% Triton and blocking with 2% goat serum (Gibco, 16210064) and 1% bovine serum albumin (BSA), sections were incubated for 1 hr with the following primary antibodies: SOX2 1:100 (Abcam, ab93689), PAX6 1:100 (Sigma-Aldrich, AMAb91372), βIII-tubulin 1:500 (Sigma-Aldrich, T8578), TBR1 1:100 (Abcam, ab31940), PSD95 1:500 (Sigma-Aldrich, MABN68), NeuN 1:100 (Sigma-Aldrich, MAB377), MAP2 1:100 (Sigma-Aldrich, ZRB2290), GFAP 1:200 (Sigma-Aldrich, G3893). After PBS washes, DAPI (100–1000 ng/ml) and secondary antibodies labeled with Alexa Fluor 488 or 594 1:500 (Invitrogen, #A-11037, #A-11029) were incubated for 1 hr. Slides were mounted using Fluoromount Aqueous Mounting Medium (Sigma Aldrich). Images were acquired using a microscope Axio Observer Z1 (Carl Zeiss), Objective 10 X, and analyzed using Fiji software.

## RNA extraction, RT-PCR

Total RNA was isolated from brain organoids with RNeasy Plus Universal Tissue Mini kit (Qiagen) and Precellys Evolution tissue homogenizer (Bertin). The concentration and purity of RNA samples were checked using the NanoDrop ND-1000 spectrophotometer at 260 and 280 nm (NanoDrop Technologies); the A260/280 ratio ranged from 1.8 to 2.2. One µg of total RNA was converted to cDNA using random primers in a volume of 20 µl using an RT2 HT first stand kit (Qiagen) according to the manufacturer's protocol. cDNA was diluted with sterile water to a final volume of 200 µL. Quantitative expression of markers were determined using 2 µl of diluted cDNA for each primer (*Supplementary file 8*) set at 10 µM using iQ SYBR Green Supermix (Biorad) in a final volume of 12 µl. Thermocycling was carried out in the CFX96 RT-PCR detection system (Bio-Rad) using SYBR green fluorescence detection. The amplification cycle used was as follows: 10 min at 95 °C, followed by 40 amplification cycles at 95 °C for 15 s, 60 °C for 60 s, and 72 °C for 30 s to reinitialize the cycle again. The specificity of each reaction was also assessed by melting curve analysis to ensure primer specificity. Relative gene expression values were calculated as $2^{-\Delta CT}$, where $\Delta CT$ is the difference between the cycle threshold (CT) values for genes of interest and housekeeping gene (GAPDH).

## Proteolysis and mass spectrometry analysis

Total pelleted cell material was supplemented with lithium dodecyl sulfate lysis buffer (Invitrogen) following the ratio of 25 µl of buffer per mg of material. Glass and silica beads (200 mg) were added to each sample, which were then subjected to 10 cycles of bead beating at 10,000 rpm for 30 s followed by 30 s halt with a Precellys device (Bertin technologies) as previously described (*Hayoun et al., 2019*). The supernatants were transferred to a new tube and incubated at 99 °C for 5 min. A volume of 20 µl of each sample was loaded on a NuPAGE 4–12% Bis-Tris gel. Then, the proteins were separated by a short electrophoresis migration (5 min) at 200 V with MES/SDS 1 X (Invitrogen) as a running buffer. Gels were stained for 5 min with SimplyBlue SafeStain (Thermo) and then washed overnight with water with gentle agitation. The polyacrylamide band containing the whole proteome from each sample was excised, treated, and proteolyzed with trypsin as recommended (*Rubiano-Labrador et al., 2014*). Peptides (300 ng) were analyzed by liquid chromatography–tandem mass spectrometry (LC–MS/MS) using an Ultimate 3000 nano-LC system coupled to a Q-Exactive HF mass spectrometer (Thermo Scientific) operated essentially as described previously (*Klein et al., 2016*) Peptides were eluted from the reverse phase chromatography column for 90 min with an acetonitrile gradient and the mass spectrometer was operated in Top20 mode, with a 10 s dynamic exclusion time for the analysis of fragmented peptides.

## MS/MS spectra interpretation and differential proteomics

MS/MS spectra were assigned using the Mascot software version 2.6.1 (Matrix Science) and the *Homo sapiens* SwissProt database. The maximum of missed cleavages, primary ions tolerance, and secondary ions tolerance were set at 2, 5 ppm, and 0.02 Da, respectively. Carbamidomethylation of cysteine and oxidation of methionine were considered as fixed and variable modifications, respectively. Peptides were identified at a p-value below 0.05 and proteins were selected when at least two distinct peptides were identified (false discovery rate below 1%). Mass spectrometry proteomics data corresponding to the 16 nanoLC-MS/MS runs are available from the ProteomeXchange Consortium via the PRIDE partner repository (https://www.ebi.ac.uk/pride/) under dataset identifiers PXD040185 and 10.6019/PXD040185.

## Bioinformatics analysis of proteomic data

### Raw data normalization and filtering

To identify the proteins differentially abundant between healthy and IPSC-CTD-derived brain organoids, raw proteomic data were normalized and filtered using an in-house script constructed using R programming language. Briefly, the Variance Stabilizing Normalization (VSN) (*Motakis et al., 2006*) was applied to normalize the data and remove the background noise. An unsupervised variation filter was then applied where any four samples with MS/MS spectral count detected were included (*Hamoudi et al., 2010*).

### Proteomics data analysis to identify the list of differentially expressed proteins between healthy and CTD brain organoids

The differential expression analysis of the proteins between healthy and CTD brain organoids was carried out using a modification of the R package for Reproducibility-Optimized Test Statistic (ROTS) to eliminate the bias in the data (*Suomi et al., 2017*). Differential expression was assessed between the following three pairs of conditions: Healthy (BJ) vs brain organoids from CTD patient (CTD1_4); Healthy (BJ) vs brain organoids from CTD patient (CTD2_3); Healthy (BJ) vs brain organoid from CTD patient (CTD3_7). The data was sorted according to *p-value* based on the Fold Discovery Rate (FDR). The differentially expressed proteins were selected by *p-value* <0.05 and FDR ≤0.25. The quality of the data separation between the various groups was assessed using Reproducibility plots and Principal Component Analysis (PCA). The identified differentially expressed proteins were visualized using volcano plots and heatmaps. The heatmaps were generated using unsupervised hierarchical clustering carried out with Ward linkage and Euclidean distance measure to assess the degree of proteomic profile separation between the different groups. The flowchart of the entire bioinformatics analysis workflow is shown in *Figure 3—figure supplement 1*.

## Pathways analysis using absolute GSEA

Absolute GSEA was adapted to identify the cellular activated pathways using enriched proteins in CTD brain organoids compared to healthy brain organoids. The GSEA was carried out as described in *Hamoudi et al., 2010*; *Harati et al., 2021* searching through more than 10,000 different cellular pathways obtained across well-annotated gene sets (c2_cp; c3_tft; c4_cgn; c5_bp; c5_mf; c7) obtained from the Broad's Institute database (https://www.gsea-msigdb.org). The pathways identified through the GSEA were sorted according to the nominal *p-value* (<0.05) as previously described (*Hamoudi et al., 2010*; *Subramanian et al., 2005*). The significantly enriched pathways in CTD brain organoids in comparison with health organoids were selected based on *p-value* <0.05. The list of enriched genes was then identified for each significant pathway and their recurrence in other pathways among all studied gene sets was searched as previously described (*Hamoudi et al., 2010*; *Harati et al., 2021*). The genes with the highest frequency across the multiple significant pathways (top 90 percentile cut-off; a total of 142 proteins) were selected for subsequent analysis.

## Identification of key proteins associated with key features of CTD patients

The GSEA and frequency analysis (top 90 percentile cut-off) identified 142 differentially expressed proteins occurring frequently across all enriched pathways. To further reduce the set of available proteins, the most abundant proteins with spectral count higher than 10 were selected, and the fold change CTD brain organoids in comparison with healthy brain organoids was determined. Proteins with fold change ≥3 were considered as upregulated and fold change ≤–3 as downregulated. This abundance and fold change analysis identified 32 abundant proteins as significantly up- or downregulated in CTD brain organoids in comparison with health organoids and in potential relation with cognitive functions. In order to identify potential functions of the differentially expressed proteins, the 32 proteins significantly altered in CTD brain organoids in comparison with health organoids were subjected to a disease-related pathway analysis carried out using Enrichr followed by a frequency analysis (*Chen et al., 2013*; *Kuleshov et al., 2016*) focusing on the following sets: BioCarta_2016, Elsevier_Pathway_Collection, GO_Biological_Process_2021, GO_Molecular_Function_2021, KEGG_2021_Human, MSigDB_Hallmark_2020, WikiPathways_2021_Human, ClinVar_2019, DisGeNET, Jensen_DISEASES, OMIM_Disease. Relevant pathways are selected based on a p<0.05 cut-off.

## Statistical analysis of the differentially abundant proteomics data

In order to identify the patterns of differentially abundant proteins, the data identified by ENRICHR analysis was used to construct a multivariate statistical model using one-way ANOVA followed by Bonferroni's post hoc test for comparisons between the healthy and CTD brain organoids.

## Western blotting

Western blotting was used to detect the abundance of GSK3β, pSer9-GSK3β, SOX2, PAK1, and MAP1B. Briefly, brain organoids were homogenized in freshly prepared lysis buffer of TBS 1 X (Biorad) supplemented with 1% Triton X-100, Protease Inhibitor Cocktail 1 X (cOmplete, Roche) and a mix of anti-phosphatase inhibitors 1 X (ammonium molybdate, sodium glycerophosphate, sodium fluoride, sodium pyrophosphate, sodium orthovanadate) using a Precellys Evolution tissue homogenizer (Bertin). The samples were then centrifuged at 10,000 g for 20 min to produce lysate for electrophoresis. 10 µg of proteins in Laemmli buffer and protein standard were loaded on 4–15% Criterion TGX Stain-Free protein gel in running buffer TGS 1 X (all from Bio-Rad) and transferred to a 0.2 µm PVDF membrane with the Trans-Blot Turbo RTA Midi Transfer Kit (Bio-Rad). Membranes were blocked for 30 min in 5% low-fat milk in TBS-Tween 20 0.1% at room temperature. Blots were probed overnight at 4 °C with the following specific primary antibodies: GSK3β 1:1000 (Cell Signaling, #9336), pSer9-GSK3β 1:500 (BD Biosciences, 610201), SOX2 1:200 (Abcam, ab93689), PAK1 1:1000 (Cell signaling, #2602), MAP1B 1:1000 (Sigma-Aldrich, M4525), and α-tubulin 1:4000 (Sigma-Aldrich, T6199). Primary antibodies were detected by horseradish peroxidase secondary antibodies diluted 1:10,000 in 5% low-fat milk in TBS-Tween 20 0.1% at room temperature. For protein detection, membranes were exposed to the Immobilon Crescendo or Forte Western HRP substrate (Millipore) in a chemidoc touch imaging system for a measurable exposure time (Bio-Rad) and quantified with Image Lab Software (Bio-Rad).

## Statistics

Statistical analysis was performed using the GraphPad Prism 9.3 program. Experimental comparisons with multiple groups were analyzed using one-way ANOVA with the Tukey's or Dunnett's multiple comparison test for post hoc analysis. For the comparison of two groups, a two-tailed Student's test was performed. A p-value of 0.05 or less was considered significant.

## Acknowledgements

This study was supported by the association Xtraordinaire and the Association for Creatine Deficiencies (ACD) as well as the CEA. The authors thank Mélodie Kielbasa, Guylaine Miotello, and Jean-Charles Gaillard (CEA) for their expert help with proteomic sample preparation and tandem mass spectrometry. The authors thank Iris Lemeunier (CEA) and Rafika Jarray (Sup'Biotech) for the preliminary experiment on CTD iPSC, and Elisa Bardou (CEA) for imaging. We are grateful to the patients included in this study.

## Additional information

### Funding

| Funder | Grant reference number | Author |
| --- | --- | --- |
| ASPIRE | | Rifat Hamoudi |
| X-traordinaire Association and CEA | | Aloïse Mabondzo |

The funders had no role in study design, data collection and interpretation, or the decision to submit the work for publication.

### Author contributions

Léa Broca-Brisson, Data curation, Investigation, Methodology, Writing – original draft, Writing – review and editing; Rania Harati, Software, Formal analysis, Investigation, Writing – original draft, Writing – review and editing; Clémence Disdier, Formal analysis, Methodology, Writing – original draft, Writing – review and editing; Orsolya Mozner, Narciso Costa, Anne-Cécile Guyot, Agota Apati, Frank Yates, Investigation, Methodology, Writing – review and editing; Romane Gaston-Breton, Auriane Maïza, Investigation, Writing – review and editing; Balazs Sarkadi, Lucie Madrange, Formal analysis, Methodology, Writing – review and editing; Matthew R Skelton, Formal analysis, Investigation, Methodology, Writing – review and editing; Jean Armengaud, Data curation, Formal analysis, Investigation, Methodology, Writing – review and editing; Rifat Hamoudi, Data curation, Software, Formal analysis, Funding acquisition, Validation, Methodology, Writing – original draft, Writing – review and editing; Aloïse Mabondzo, Conceptualization, Data curation, Formal analysis, Supervision, Funding acquisition, Investigation, Visualization, Writing – original draft, Project administration, Writing – review and editing

### Author ORCIDs

Orsolya Mozner (iD) http://orcid.org/0000-0001-5784-7702
Balazs Sarkadi (iD) http://orcid.org/0000-0003-0592-4539
Agota Apati (iD) http://orcid.org/0000-0003-0380-8139
Jean Armengaud (iD) http://orcid.org/0000-0003-1589-445X
Aloïse Mabondzo (iD) https://orcid.org/0000-0002-0627-8949

### Ethics

BJ primary fibroblasts were obtained from ATCC (CRL-2522). SP fibroblasts were isolated from normal placenta according to the protocol study approved by the local ethics committee (Advisory Committee for the Protection of Persons in Biomedical Research Cochin Hospital, Paris, n°18-05). Human fibroblasts from CTD patients, were obtained from skin biopsy specimens and were a gift from the Centre de Référence des Maladies Héréditaires du Métabolisme at the Necker Hospital in Paris. Three patients with cerebral creatine deficiency caused by lack of creatine transporter were studied.

Informed and a written consent was obtained from all anonymized human CTD subjects, and experiments were carried out in accordance with relevant guidelines and regulations. All the mutations were previously described by Valayannopoulos et al., 2013.

Reviewer #1 (Public Review): https://doi.org/10.7554/eLife.88459.3.sa1
Reviewer #2 (Public Review): https://doi.org/10.7554/eLife.88459.3.sa2
Author Response https://doi.org/10.7554/eLife.88459.3.sa3

## Additional files

### Supplementary files
• Supplementary file 1. Raw proteomic Data showing the list of proteins and their spectral counts generated by High-resolution tandem mass spectrometry on the 16 biological samples generated a very large dataset comprising a total of 943,656 MS/MS spectra and the monitoring of the abundance of 4219 proteins.

• Supplementary file 2. Normalized and filtered proteomic data. The raw proteomic data consisting of 4219 proteins were normalized using Variance Stabilizing Normalization (VSN) and filtered using an unsupervised variation filter.

• Supplementary file 3. Normalized and filtered proteomic data were sorted according to p-value (p<0.05 and logfc). The differential expression analysis of the proteins between healthy and CTD-derived brain organoids was carried out using a modification of the R package for Reproducibility-Optimized Test Statistic (ROTS). The data was sorted according to *p-value* (<0.05) based on FDR ≤0.25. The blue color indicated proteins downregulated in CTD compared to BJ organoids. The red color indicates proteins upregulated in healthy (BJ) vs CTD-derived brain organoids.

• Supplementary file 4. Frequency of proteins identified by GSEA. Absolute GSEA was carried out searching through more than 10,000 different cellular pathways obtained across well-annotated gene sets (c2_cp; c3_tft; c4_cgn; c5_bp; c5_mf; c7). The pathways identified through the GSEA were sorted according to the nominal *p-value* (<0.05). The list of enriched genes was then identified for each significant pathway and their recurrence in other pathways among all studied gene sets. The genes with the highest frequency across the multiple significant pathways (top 90 percentile cut-off; a total of 142 proteins) were selected for subsequent analysis.

• Supplementary file 5. List of 142 proteins sorted according to the fold change. The 142 identified by the GSEA and frequency analysis (top 90 percentile cut-off) as differentially expressed proteins occurring frequently across all enriched pathways were sorted according to fold change; the most abundant proteins with fold change (–3<fold change >+3) (48 proteins, highlighted in yellow) and a sc higher than 10 (32 proteins, highlighted in dark yellow) were selected for further analysis. The 32 proteins selected are the following: RUFY3; MAP1B; GSK3B; PAK1; STMN1; FYN; GNAI1; MYO5A; PAK2; DCTN1; LMNA; SRC; GNAQ; CDK5; PLCG1; MECP2; ACTN3; HMGB2; SYT1; CALM1; ANXA2; L1CAM; ANK2; ACTN2; TNNT2; MYH6; ANXA1; PTK7; TPM1; SFRP1; POSTN; CASQ2.

• Supplementary file 6. Frequency of 32 proteins identified by Enrichr analysis. An ENRICHR analysis was performed on the 32 proteins to identify the proteins in potential relation with the cognitive functions. The pathways with p-value less than 0.05 from the following libraries: (BioCarta, Elsevier_ Pathway_Collection, GO_Biological_Process, GO_Molecular_Function, KEGG_Human, MSigDB_ Hallmark, WikiPathways_Human). The diseases with p value less than 0.05 from the following libraries: (ClinVar, DisGeNET, Jensen_DISEASES, OMIM_Disease).

• Supplementary file 7. 32 proteins functions and relations to creatine transporter deficiency (CTD) symptoms.

• Supplementary file 8. Primer sequences used for real-time PCR (RT-PCR) analysis.

• MDAR checklist

### Data availability
Mass spectrometry proteomics data corresponding to the 16 nanoLC-MS/MS runs were deposited at the ProteomeXchange Consortium via the PRIDE partner repository (https://www.ebi.ac.uk/pride/) under dataset identifiers PXD040185 and https://doi.org/10.6019/PXD040185.

The following dataset was generated:

| Author(s) | Year | Dataset title | Dataset URL | Database and Identifier |
|---|---|---|---|---|
| Broca-Brisson et al | 2023 | Proteomes of human brain organoids derived from individuals with creatine transporter deficiency | https://doi.org/10.6019/PXD040185 | PRIDE, 10.6019/PXD040185 |

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
