## [Editor Report · eLife assessment]

This is an **important** study highlighting how a single protein transporter dysfunction can significantly alter brain biochemistry, potentially playing a crucial role in the intellectual disability in creatine transporter deficiency (CTD) patients. The evidence is **compelling** that the new in vitro CTD model using CTD patient's brain organoid cultures will be widely applicable. Despite minor areas for further exploration, the study significantly enhances our understanding of CTD, offering potential therapeutic targets and a robust foundation for continued research in the field.

---

## [Referee Report · Reviewer #1 (Public Review)]

This well written and designed study by Broca-Brisson et al describes the generation of a new in vitro model for creatine transporter deficiency (CTD), making use of human brain organoid cultures derived from CTD patients. This new model will certainly prove itself very useful to better understand this genetic disease essentially affecting CNS. As CTD has no satisfactory treatment so far (despite more than 20 years of research), this new model will also be very useful to design and develop new treatments.

In particular, through the use of immunohistochemistry, real time PCR, and proteomics combined with integrative bioinformatic and statistical analysis, authors provide very interesting new information on the brain pathways affected in CTD (e.g. neurogenesis with down-regulation of SOX2 and PAX6 but up-regulation of GSK3b; and proteins involved in autistic spectrum, epilepsies or intellectual disabilities).

---

## [Referee Report · Reviewer #2 (Public Review)]

In their recent manuscript, Broca-Brisson et al. deliver a multidisciplinary approach to investigate creatine transporter deficiency (CTD) using human-derived brain organoids. The authors have provided a compelling CTD human brain organoid model using induced pluripotent stem cells (iPSCs) derived from individuals with CTD. This model shows distinct differences in creatine uptake between organoids originating from CTD patients and their healthy counterparts. Furthermore, the researchers effectively restored creatine uptake by reintroducing the wild-type CRT in the iPSCs.

The team used advanced molecular biology techniques and sophisticated mass spectrometry to identify changes in protein regulation within these CTD brain organoids. They propose an intriguing theory linking reduced creatine uptake to abnormalities in the GSK3β kinase pathway and mitochondrial function, which might underlie intellectual disability seen in CTD patients.

This study is well-structured and easy to follow, with clear and concise explanations of the experiments. The authors present an important idea: a dysfunction in just one protein transporter (CRT) can cause significant biochemical changes in the brain. Their findings are well-presented and backed by high-quality figures and comprehensive data analysis.

There are only minor suggestions for improvement in this manuscript. The authors strongly link creatine uptake, the GSK3β pathway, and intellectual disability. Enhancing this claim with data on phosphorylation differences between organoids derived from healthy individuals and those from CTD patients could solidify this foundation and facilitate a more holistic understanding of the disease. In addition, the in vitro model based on organoids might be closer than other experimental setups; however, proving that those differences are also present in vivo would greatly benefit the story.

There is also some uncertainty around the rescue experiment using the exogenous SLC6A8 gene. Could the difference in creatine uptake between the rescue iPSCs and the healthy control be due to CRT overexpression? Higher levels of the transporter may explain the elevated levels of intracellular creatine. Thus, a comparison using Western blotting experiments could be a valuable addition to evaluating the expression levels of this protein.

Overall, this study provides valuable insights into CTD and potential therapeutic targets. It enriches our understanding of CTD and opens up new avenues for future research in this field.

---

## [Author Response]

The following is the authors’ response to the original reviews.

**Reviewer 1**
Question 1: While the CTD human brain organoids show a decrease in Cr (in absence of Cr in the culture medium) as compared to control organoids (4 times less), they are not devoid of Cr. Do these organoids express the two enzymes allowing Cr synthesis (AGAT and GAMT), and in which brain cell types? If yes, how to explain the decrease in Cr in the CTD organoids?

There is a lack of functional CRT in the CTD human brain organoids. The basal level of creatine in CTD human brain organoid is significantly lower than in healthy human brain organoids. The intracerebral creatine synthesis is due to different expression of the AGAT and GAMT enzymes and relies on functional CRT for the transport of the GAA intermediate. Literature pointed out that both enzymes are rarely co-expressed (Braissant et al., 2001, PMID: 11165387) meaning that GAA intermediate needs to be transported by CRT to neurons for complete creatine synthesis. Even if we evidenced a slight mRNA expression of AGAT and GAMT enzymes, the creatine synthesis is not effective since the GAA intermediate could not be transported in cell expressing GAMT due to the non-functional creatine transporter in the CTD human brain organoids.

Question 2. The rescue experiment, re-establishing a functional Cr transporter (CRT or SLC6A8) in the CTD human brain organoids, is very interesting, as this may help the design and development of new treatments for CTD. However, authors claim that the functional CRT expressed in the rescued CTD organoids was expressed in each cell. This may be a difficulty in the development of new CTD treatments, as CRT should be expressed in neurons and oligodendrocytes, but not in astrocytes. Authors may want to comment on this point.

As shown in Figure S2C, the whole brain organoid in the rescue experiment shows the expression of the GFP protein, thus also the co-expressed wild-type CRT. In these experiments, we did not make a detailed cellular characterization of the rescued organoids, and this may be the aim of a separate study that will carry out experiments for an exact characterization of the cell-specific CRT expression and function in the rescued brain organoids. Accordingly, we corrected in the revised version of manuscript the statement on page 6 to the following: “SLC6A8 expressing brain organoids showed GFP fluorescence in the whole area of the organoid (Fig S2C).”

**Reviewer #1 (Recommendations for The Authors):**
Authors may cite the recent review by Fernandes-Pires (2022) exposing the challenges to treat CTD (introduction, lines 57-58 for example).

Reference has been added, lines 57-58 of the revised version

Authors may precise in their introduction (lines 60-61) that, while creatine (Cr) supplementation is not effective to treat CTD male patients, a proportion of female CTD patients is responsive to Cr supplementation (due to the differential inactivation of one of the X chromosome depending on the cells).

Treating CTD appears simple: transport creatine into the brain cells. In individuals with creatine synthesis disorders, increasing brain creatine levels thanks to oral supplementation of creatine monohydrate and/or precursors improves neurodevelopmental outcomes. This task has proven more daunting than expected in CTD since oral creatine supplementation does not increase brain creatine concentrations. Literature and more specially data reported by Van de Kamp “X-linked creatine transporter deficiency: clinical aspects and pathophysiology. J Inhert Metab Dis 37 (5):715-733 describes 3 females CTD patients without improvement of clinical outcomes. Bruun et al., 2018 “Treatment outcome of creatine transporter deficiency: international restrospective cohort study: Metab. Brain Dis: 33:875-884 reports 2/3 CTD females with improvement of clinical outcome. Taken together the sentence has been modified in the revised version of the manuscript as follows: “Several combinations of nutritional supplements or Cr precursors l-arginine and l-glycine, have been studied as therapeutic approaches for CTD, but they have shown limited success (Bruun et al., 2018, Valayannopoulos et al., 2013) (lines 61-63, Page 4)

When comparing their new in vitro CTD model of human brain organoids with existing in vivo rodent models, authors may add the citation of the rat model of Duran-Trio et al (2021 & 2022), in particular for its description of CNS tissue alterations (dendritic spines density for example).

The reference Duran-Trio et al (2021) has been added (page 4, line 70). The reference Duran-Trio et al (2022) has been added (page 11) and the sentence has been modified in the revised version of the manuscript as follows: “Reduced cortical spine density and reductions in protein levels of several synaptic markers have been observed in the brains of Slc6a8-/y mice and rats (Chen et al., 2021; Duran-Trio et al., 2022)”.

**Reviewer #2 (Recommendations For The Authors):**
There are only minor suggestions for improvement in this manuscript. The authors strongly link creatine uptake, the GSK3β pathway, and intellectual disability. Enhancing this claim with data on phosphorylation differences between organoids derived from healthy individuals and those from CTD patients could solidify this foundation and facilitate a more holistic understanding of the disease. In addition, the in vitro model based on organoids might be closer than other experimental setups; however, proving that those differences are also present in vivo would greatly benefit the story.

As shown in Fig 6A-B, GSK3β is less phosphorylated on Ser9 in CTD brain organoids compared to healthy organoids, indicating that GSK3β is more active in organoids with reduced creatine levels. Studying the level of GSK3β phosphorylation in the mouse brain could be part of next experiments and another story.

There is also some uncertainty around the rescue experiment using the exogenous SLC6A8 gene. Could the difference in creatine uptake between the rescue iPSCs and the healthy control be due to CRT overexpression? Higher levels of the transporter may explain the elevated levels of intracellular creatine. Thus, a comparison using Western blotting experiments could be a valuable addition to evaluating the expression levels of this protein.

For the rescue experiment, we used a vector where SLC6A8 and eGFP were connected by an IRES2 sequence, providing simultaneous, but independent expression of the two proteins. CTD-rescue iPSC clones were selected based on high eGFP fluorescence. These clones probably have several copies of transgene in their genome, which could result in a higher abundance of SLC6A8 compared with healthy iPSCs. The difference in creatine uptake between the CTD-rescue iPSCs and the healthy control is probably due to CRT overexpression. However, there are no satisfactory anti-SLC6A8 antibodies commercially available to quantify CRT by western-blot. We would like to add that, although creatine uptake is higher in CTD-rescue iPSCs than in healthy control, the basal level of creatine (which corresponds to culture conditions for the rest of the experiments) is similar.

Overall, this study provides valuable insights into CTD and potential therapeutic targets. It enriches our understanding of CTD and opens up new avenues for future research in this field.

We thank the reviewer for their kind words and hope this study will be useful for other researchers in the CTD field.